# Influence of the new standardized clinical cryopreservation/slow thawing protocol on immunogenicity of arterial allografts in rats

Jan Hruby[1], Rudolf Spunda[1], Pavel Mericka[2], Mikulas Mlcek[3], Ondrej Pecha[4], Katrin Splith[5,6], Moritz Schmelzle[5,6], Felix Krenzien[5,6], Jaroslav Lindner[1], Miroslav Spacek[1‡], Ivan Matia[7,8‡]*

1 2nd Department of Surgery–Department of Cardiovascular Surgery, 1st Faculty of Medicine, Charles University in Prague and General University Hospital in Prague, Czech Republic, 2 Tissue Bank, Faculty Hospital Hradec Kralove, Charles University- Faculty of Medicine in Hradec Kralove, Hradec Kralove, Czech Republic, 3 Institute of Physiology, First Faculty of Medicine, Charles University in Prague, 4 Technology Centre of the Academy of Sciences of the Czech Republic, Prague, Czech Republic, 5 Department of Surgery, Campus Charité Mitte and Campus Virchow-Klinikum, Charité-Universitätsmedizin Berlin, Berlin, Germany, 6 Translational Centre for Regenerative Medicine, University of Leipzig, Leipzig, Germany, 7 Department of Cardio-Vascular Surgery, Nord Hospital and Karl Landsteiner Institute for Cardio-Vascular Research, Vienna, Austria, 8 Teaching Center, Medizinische Universität Wien, Vienna, Austria

‡ Joint Senior Authors
* ivanmatia@gmail.com

**Data Availability Statement:** All relevant data are within the paper and its Supporting Information files.

## Abstract

### Objectives and design

At the present time there are two waiting list for patients with vascular prosthetic infection indicated for arterial transplantation in the Czech Republic. The inclusion of each patient for cold-stored or cryopreserved arterial transplantation is the preference of indicating surgeon. In this experimental work we studied the immunogenicity of rat aortal allografts treated by our new clinical cryopreservation/slow thawing protocol.

### Material and methods

Brown-Norway (BN) (N = 6, 203–217 g) or Lewis (LEW) (N = 6, 248–254 g) abdominal aortal grafts treated in accordance with our new clinical cryopreservation/slow thawing protocol were orthotopically transplanted to Lewis recipients (N = 12, 191–245 g). Aortal wall histology and infiltration by recipient immune cells, as well as donor specific anti MHC class I and II antibodies in recipient serum were studied in both isografts and allografts on day 30 post-transplant. Core data of cryopreserved allografts were compared to our previous data of cold-stored aortal allografts treated in accordance with our clinical cold-storage protocol.

### Results

Cryopreserved allografts showed regular morphology of aortal wall with clear differentiation of all three basic anatomical layers on day 30 postransplant. Intimal layer showed no hyperplasia, luminal surface was covered by endothelial cells. No statistical difference was observed in tunica media thickness between isografts and allografts. The medial layer

**Funding:** The authors received no specific funding for this work.

**Competing interests:** The authors have declared that no competing interests exist.

showed no necrosis, shrinkage or immunoglobuline G deposition in any experimental group. The adventitial infiltration by immune cells was significantly higher (P<0.05) in allografts. Cryopreserved allografts showed significant lower activation of both cell- and antibody mediated immunity compared to historical data of cold-stored allografts.

## Conclusion

Aortal wall histology of rat allografts treated by our new standardized clinical cryopreservation/slow thawing protocol was comparable to that of the cryopreserved isografts on day 30 posttranspant. The immunogenicity of cryopreserved aortal allografts was significantly lower compared to that of cold-stored aortal allografts.

## Introduction

Incidence rates of aortic grafts infection in patients after primary aortic surgery in the current endovascular era is considerable [1]. Major aortic graft infection is a life-threatening complication with high mortality and morbidity rates [2,3]. One of the most effective treatment modality of this devastating complication is the reoperation with replacement of an infected prosthesis by a cold-stored [4] or cryopreserved arterial allograft [5].

The recovery techniques of arterial allografts and subsequent cold-stored or cryopreservation/thawing protocols published by large vascular centers are very inhomogeneous [6,7]. The different properties of storage solutions, variation in cold ischemic time prior to implantation (cold-stored allografts) or cryopreservation, different freezing and thawing protocols are used worldwide [4–6]. All these aspects can influence the final quality of implanted arterial allografts and caused significant differences in the early and late graft-related complications [6].

In the Czech Republic, the cold stored arterial allografts are used as a substitute for infected vascular prosthesis since middle 1990s [7]. Both solid organ and cold-stored arterial transplantation programs are governed by the same legislation and managed by the "Transplantation Coordination Centre of the Czech Republic". Therefore, the cold-stored arteries obtained during multi-organ recovery from braindead donors are stored in organ conservation solutions and must be transplanted to the patient included in the special "waiting list" within 48 hours after recovery [8].

However, cold-stored allografts were not in sufficient abundance for all the patients on the waiting list. Therefore, the clinical cryopreserved arterial transplantation program was started in 2011. All of arterial allografts included to the national program are processed exclusively in the Tissue Bank of The University Hospital Hradec Kralove under new standardized cryopreservation/slow thawing protocol [9].

At the present time there are two waiting list for patients indicated for arterial transplantation in the Czech Republic. The inclusion of each patient for cold-stored or cryopreserved arterial waiting list is the preference of indicating surgeon [9].

The experimental and clinical studies revealed strong antigenicity of both cold-stored and cryopreserved allografts, respectively [10,11]. This immunogenicity is responsible for late graft-related complication presented as graft dilatation, rupture or thrombosis [12]. In addition, this immunogenicity is influenced by all procedure steps in their pretransplant history [10].

Therefore the aim of our study was:

1. to transfer all steps of the new national standardized clinical cryopreservation/slow thawing protocol to the experimental settings,

2. to study acute immune reaction of rats recipients of cryopreserved arterial grafts treated by this protocol,

3. to compare the data of current cryopreservation/slow thawing experiment to data obtained previously by our cold-storage rat experiment to study the differences in immunogenicity of both cryopreserved and cold-stored aortal allografts, respectively.

## Material and methods

### Ethics statement

Principles of laboratory animal care were followed, and applicable national laws observed during the study. The study protocol was approved by the Committee on 1. Medical faculty of Charles University in Prague (1.LF 563/13, MSMT-14808/2014-6).

### Animals

Male Brown-Norway rats (BN) ($RT1^n$) (N = 3) weighting 203–217 g were used as donors of abdominal aortal allografts, and male Lewis rats (LEW) ($RT1^l$) (N = 3) weighting 248–254 g were used as donors of abdominal aortal isografts, respectively. The recipient animals of cryopreserved aortal grafts (male Lewis rats (LEW) ($RT1^l$) were divided into two groups. Group CRYO-ISO was that of cryopreserved isografts (LEW to LEW, N = 6, weighting 191–250 g), group CRYO-ALLO was that of cryopreserved allografts (BN to LEW, N = 6, weighting 193–245 g).

Animals were obtained from Charles River, Germany and maintained according to the National Institute of Health guidelines. Each transplanted animal was housed in a separate cage during the entire 30-day follow-up period.

### Operative procedure

All aspects of orthotopic abdominal aortic transplantation was described in details in our previous publications [13,14]. The cryopreserved aortal grafts in recent experiment were transplanted by the same anesthesiological and surgical techniques used previously in our cold-stored aortal grafts experiment. Both experiments were supervised by the same principal investigator (I.M.). Briefly, a 2–2.5 cm long segment of the infrarenal aorta was excised, prepared and stored in accordance to cryopreservation protocol. Cryopreserved aortal grafts were implanted orthotopically into the recipient's infrarenal aorta after a midline laparotomy using a 10/0 mono-filament interrupted suture. No anticoagulants or anti-platelet drugs were used in recipients during the 30-day follow-up period.

### Protocol of cryopreservation

The new clinical cryopreservation/slow thawing protocol was modified to experimental operating room conditions of Institute of Physiology of $1^{st}$ Medical faculty at the Charles University in Prague.

The all donor animals of each experimental group (CRYO-ISO (N = 3) and CRYO-ALLO (N = 3) were operated at once. Each abdominal aortal graft was flushed after the recovery with 2 ml of Custodiol solution (Custodiol®, Dr. Franz Köhler Chemie GmbH, Germany) containing 100 IU/ml of heparin and placed into 10 ml of pre-cooled Custodiol solution, thereafter. Each graft was stored at the temperature of melting ice in closed sterile certified plastic jars

(Medfor 250 ml Farnborough, UK) until finalization of all donor´s operations. Thereafter, the aortal grafts were put into double sterile disposable plastic bags (CryoMACS Freezing Bag 500, Miltenyi Biotec GmbH, Germany) containing 25ml of pre-cooled 6% solution of hydroxyethylstarch m.w. 130.000 Da (Voluven 6%, Fresenius Kabi, Germany). No antibiotics were added. The content of the bag was subsequently mixed with pre-cooled 20% dimethylsulfoxide (DMSO) (WAK ChemieMedical GmbH, Germany) in the 6% hydroxyethylstarch solution (Voluven 6%, Fresenius Kabi, Germany) and each bag was closed by sealing at the sealing machine (STERISEAL B 83-R, Cevor s.r.o., Troubsko u Brna, Czech Republic). The bags were put into outer metal cassette (ST 100, Consarctic GmbH, Schölkrippen, Germany) and stored at the temperature of melting ice until the beginning of freezing process. The controlled–rate freezing by the rate of -1K/min to -90˚C and -5K/min to -150˚C was performed at the programmable freezer (Kryo-10, Planer Biomed, Sunbury on Thames, England). After completion of freezing process, the cassettes were transferred in liquid nitrogen vapour phase to the Tissue Establishment of the Institute of Haematology in Prague. The grafts were stored in this institution in liquid nitrogen vapour phase by temperature of -190˚C until the day of implantation.

## Protocol of thawing

The cassettes with aortal grafts were transported at the day of implantation in liquid nitrogen vapour phase from the Tissue Establishment to the operating room of Institute of Physiology. The cassettes were removed from the shipper in the operating room and placed for about 60 minutes into a refrigerator with temperature of +4˚C and then were kept for about 30 minutes by the room temperature. Subsequently, the aortal grafts were removed from the bag and each graft was divided into two pieces of an identical length to be used in two rat recipients. After dividing each aortal graft was stored separately in 10 ml of Custodiol solution in the refrigerator with temperature of +4˚C until the beginning of anastomosis in the recipient animal. No antibiotics were added.

## Time-periods of cryopreservation and thawing

The whole process of aortal recovery, cryopreservation, storage, thawing and transplantation was divided into the six time-periods. The duration of each time-period was measured for each aortal graft separately. Detailed definitions and duration of each time–period of the cryopreservation/slow thawing protocol for each experimental group are summarized in Table 1. The individual data per aortal graft are given in S1 Table.

The total cold ischemic time of cryopreserved aortal grafts (pre-freezing + post-thawing cold ischemic time) was 313±62 minutes for isografts and 504±198 minutes for allografts, respectively. No statistical difference was observed between both groups.

## Blood samples

Blood samples for determination of donor specific anti-MHC Class I and II antibodies on day 0, and day 30 in all groups were collected by orbital sinus puncture as described previously [13].

## Aortal grafts explantation on day 30

After the blood samples collection (see above) a midline re-laparotomy was performed in anaesthetized animals. Aortal grafts were excised after the administration of heparin (100 IU/kg). The animals were then euthanized by intracaval administration of a lethal dose of thiopental [13].

**Table 1. Time-periods of the cryoconservation/slow thawing procedure.**

| | Time periods | | Cryopreserved grafts | |
|---|---|---|---|---|
| Number | Name | Definition | CRYO-ISO | CRYO-ALLO |
| 1 | PRE-FREEZING CIT (cold ischemic time) | Time period between aortal clamp in the donor animal and insertion of aortal grafts to DMSO solution. Aortal grafts were stored in Custodiol solution at the temperature of melting ice during this period. | 03:12 hours (min 02:54, max 03:28) | 03:31 hours (min 01:23, max 04:45) |
| 2 | DMSO time | Time period between insertion of aortal grafts to DMSO solution and start of cryoconservation at the programmable freezer. | 00:40 hours (min 00:38 max 00:42) | 00:38 hours (min 00:19, max 00:48) |
| 3 | CRYO time | Time period between the start of cryoconservation of aortal grafts at the programmable freezer and insertion of cassetes to liquid nitrogen | 02:38 hours | 02:38 hours |
| 4 | NITROGEN time | Time period between insertion and removal of cassetes out of liquid nitrogen. | 172,6 days (min 171, max 176) | 179,3 days (min 176, max 181) |
| 5 | THAWING time | Time period between removal of cassetes out of liquid nitrogen vapour phase and insertion of aortal grafts to Custodiol solution. | 01:33 hours (min 01:10, max 01:45) | 01:25 hours (min 01:10, max 01:33) |
| 6 | POST-THAWING CIT (cold ischemic time) | Time period between insertion of aortal grafts to Custodiol solution and reperfusion of aortal graft in recipient animal. | 02:00 hours (min 00:58, max 03:27) | 04:53 hours (min 02:26, max 07:07) |

The process of recovery, cryoconservation, storage, thawing and transplantation of cryopreserved aortal grafts was divided into six defined and measured time periods.

CRYO-ISO—cryopreserved isografts

CRYO-ALLO—cryopreserved allografts

## Parameters under study

**Histological and immunohistological analysis of explanted aortal grafts.** All histological and immunohistological techniques used were identical to those used in cold-stored aortal grafts experiment [13]. In short, the sections for histological and immunohistological analysis were taken from the midportion of the graft. The sections were stained with a Hematoxylin-Eosin and Van Gieson with elastica stain and analyzed using Olympus DP-Soft software. Medial thickness were measured up to 10 locations in each section. The mean value and standard deviation (SD) (mean ± SD) were calculated for each aorta as well as animal group. The specific antibodies for detection of endothelial cells, CD4+, CD8+, Lewis MHC class II+ cells, immunoglobulins G were used for immunohistological examination of aortal grafts. Detailed specifications of antibodies and immunohistological techniques are summarized in Table 2.

The slides were scored in a blinded fashion. CD4+, CD8+ and Lewis MHC class II+ cells were counted at 10 locations at x1000 magnification. The cellularity was defined as the mean value of the cells counted.

**Flow cytometry analysis of blood sample.** In vitro binding of cryopreserved aortal grafts recipient sera to quiescent Brown-Norway splenocytes was determined by flow cytometry. All procedures were identical to those used previously in cold-stored aortal grafts experiment [15]. Detailed specifications of antibodies used are summarized in Table 2.

**Statistical analysis.** The values in the text and tables are expressed as the mean±standard deviation (SD). Comparisons of parameters under study between experimental groups (total cold ischemic time, tunica media thickness, CD4+, CD8+, MHC II+ cells, anti MHC I and anti MHC II antibodies) were performed using the analysis of variance (ANOVA), followed by the Tukey HSD Multiple Comparisons test. All analyses were conducted in Stata (version 12.1).

## Results

### Animals

Twelve aortal grafts were successfully transplanted. All recipient animals survived the whole 30 day follow-up period.

**Table 2. Detailed specifications of antibodies and immunohistological techniques used.**

| Histology of aortal grafts | Hematoxilin eosin | |
|---|---|---|
| | Van Gieson elastic stain | |
| Immunohistology of aortal grafts | CD4+ cells | Primary antibody: anti-CD4 (W3/25, Cymbus Biotechnology LTD, Hampshire, UK) Secondary antibody: Histofine® Simple Stain Rat MAX PO (Nichirei Biosciences Inc., Japan) Detection system: Dako Liquid DAB+ Substrate-Chromogen System (Dako Denmark A/S, Glostrup, Denmark) |
| | CD8+ cells | Primary antibody: anti-CD8 (OX-8, Cymbus Biotechnology LTD, Hampshire, UK) Secondary antibody: Histofine® Simple Stain Rat MAX PO (Nichirei Biosciences Inc., Japan) Detection system: Dako Liquid DAB+ Substrate-Chromogen System (Dako Denmark A/S, Glostrup, Denmark) |
| | Endothelial cells | Primary antibody: anti-Von Willebrand factor (Dako Denmark A/S, Glostrup, Denmark) Secondary antibody: Histofine® Simple Stain Rat MAX PO (Nichirei Biosciences Inc., Japan) Detection system: Dako Liquid DAB+ Substrate-Chromogen System (Dako Denmark A/S, Glostrup, Denmark) |
| | Lewis MHC class II + cells | Primary antibody: anti RT1.B$^u$ (MRC-OX3, Cedarlane Laboratories Ltd., Burlington, Canada) Secondary antibody: horse anti mouse antibody (Vector Lab, Burlingame, USA) Detection system: R.T.U. Vectastain Elite ABC Reagent (Vector Lab, Burlingame, USA) |
| | Immunoglobulines G | Primary antibody: anti rat IgG cunjugated with fluoroscein isothianate (Chemicon International Inc.,Temecula, USA) |
| Flow cytometry analysis of aortal grafts recipients sera | MHC Class I | Primary antibody: anti-RT1.Ac (OX-27, Acris Antibodies GmbH, Herford, Germany) Secondary antibody: PE-Cy7-Streptavidin (BD Biosciences, Heidelberg, Germany) |
| | MHC Class II | Primary antibody: anti-RT1.D (OX-17, BD Biosciences, Heidelberg, Germany) Secondary antibody: PE-Cy7-Streptavidin (BD Biosciences, Heidelberg, Germany) |

Specification of monoclonal and polyclonal antibodies used for histological analysis of aortal grafts (30 days after transplantation) and flow cytometry analysis of recipients sera (pretransplant and 30 days after transplantation, respectively).

## Histology and immunohistology of cryopreserved isografts and allografts on day 30 posttransplant

The both isografts and allografts explanted on day 30 posttransplant showed regular morphology of aortal wall with clear differentiation of all three basic anatomical layers (Fig 1A–1F). The luminal surface of tunica intima was covered by a monolayer of endothelial cells. No signs of intimal hyperplasia and no Lewis MHC group II positive cells infiltration of intimal layer were detected. The medial layer of both isografts and allografts showed no signs of necrosis or immunoglobulin G depositions (Fig 1F). Indeed, no statistical difference was observed in the thickness of tunica media of both groups (Table 3). The adventitial infiltration with MHC class II+ cells of Lewis origin, CD4+ and CD8+ cells was significantly higher in aortal allografts (Fig 1D and 1E) compared to isografts (Fig 1A and 1B) (Table 3).

## Serum anti-MHC Class I and Class II antibodies in recipients of cryopreserved aortal isografts and allografts

Pretransplant as well as day 30 sera from isografted animals showed no inhibition of fluorescence-labelled MHC class I and class II antibody binding to donor quiescent splenocytes.

Day 30 sera from allografted animals showed higher inhibition of fluorescence-labelled MHC class I and class II antibody binding to donor quiescent splenocytes compared to day 0

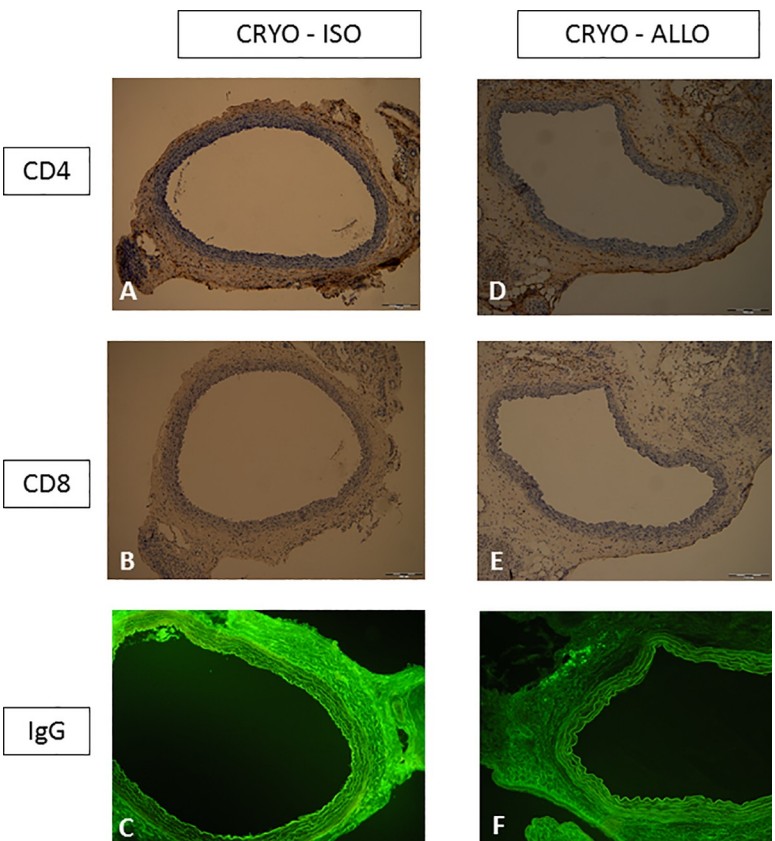

**Fig 1. a,b,c,d,e,f. Representative light microscopic histological features of both isografts (a,b,c) and allografts (d,e,f) treated by our new cryoconservation/slow thawing protocol and obtained on day 30 following transplantation.** Both isografts (Fig 1A, 1B, and 1C) and allografts (Fig 1D, 1E and 1F) showed clear differentiation of all three basic anatomical layers with no signs of intimal hyperplasia. No signs of medial layer destruction with immunoglobulins G deposition was detected in either isografts (Fig 1C) or allografts (Fig 1F). The adventitial infiltration of cryopreserved allografts by CD4+ (Fig 1D) and CD8+ cells (Fig 1E) (stained brown) was significantly higher compared to isografts (Fig 1A and 1B). Fig 1A and 1D –Anti-CD4 antibody, original magnification x 100, positive cells stained brown. Fig 1B and 1E –Anti-CD8 antibody, original magnification x 100, positive cells stained brown. Fig 1C and 1F –Anti-IgG fluorescein isothiocyanate-conjugated antibody, original magnification x 100.

sera. However statistical significance (p>0.05) was observed only by MHC class I antibodies (Table 4).

## Comparison of immunogenicity of rat aortal allografts treated by new clinical cryopreservation/slow thawing protocol to rat aortal allografts treated by clinical cold-storage protocol

The protocol used in the clinical program of cold-stored arterial allografts transplantation in the Czech Republic was modified to experimental operating room conditions and described in details in our previous publication [13]. The original core data of recent experiment were compared to historical data of our cold-stored allografts experiment [13]. Both experiments were supervised by the same principal investigator (I.M.).

The cold-stored allografts showed significant higher immunogenicity compared to cryopreserved allografts on day 30 posttranspant. The hyperplastic intima was infiltrated by recipient MHC class II+ cells and CD8+ cells. The medial layer showed signs of necrosis and deposition of immunoglobulins G and was significantly thinner (P<0.05) compared to cryografts (Fig 2).

**Table 3. Histological and immunohistological parameters of cryopreserved aortal grafts under study on day 30 posttransplant.**

| | | CRYOGRAFTS on 30 POD | |
|---|---|---|---|
| | | ISOGRAFTS (CRYO-ISO, N = 6) | ALLOGRAFTS (CRYO-ALLO, N = 6) |
| Intimal layer | Endothelial layer | + | + |
| | Intimal hyperplasia | - | - |
| Medial layer | SMC necrosis | - | - |
| | IgG deposition | - | - |
| | Medial thickness (mikrometer) | 79.3 ± 15.4 | 75.4 ± 14.9 |
| Adventitial layer | CD8+ cells | 2.2 ± 2.7 | 6.9 ± 5.4* |
| | CD4+ cells | 3.9 ± 2.6 | 9.6 ± 6.5* |
| | LEW MHC class II+ cells | 6.3 ± 4.4 | 20.7 ± 6.7* |

The rat aortal allografts processed in accordance with new clinical cryoconservation/slow thawing protocol showed normal anatomical structure of aortal wall with higher concentrations of immunocompetent cells in adventitial layer compared to isografts.

* The total amounts of CD4+, CD8+ and Lewis MHC class II+ cells in adventitial layer of cryopreserved aortal allografts were significantly higher (P<0.05) than those observed in cryopreserved aortal isografts

The adventitial layer of cold-stored allografts showed ten-fold higher infiltration by CD4+ and CD8+ cells when compared to cryopreserved allografts.

Day 30 recipient sera of both cryopreserved and cold-stored allografts showed significant higher inhibition of fluorescence-labelled MHC class I antibody binding to donor quiescent splenocytes compared to preoperative values. However, the statistically higher inhibition of fluorescence-labelled MHC class II antibody binding to donor quiescent splenocytes compared to preoperative values was observed only in recipients of cold-stored allografts.

## Discussion

The present study examined influence of the new standardized clinical cryoconservation/slow thawing protocol used in the Czech national "Cryopreserved vascular grafts program" on acute immunogenicity of cryopreserved aortal allografts in rats. This protocol allowed to process the aortal allografts by short pre-freezing cold ischemia, controlled freezing process, slow thawing process and short post-thawing cold ischemia as well. Indeed, using this protocol the aortal allografts were stored in modern solid organ conservation solution during both cold ischemic periods.

The rat cryopreserved aortal allografts showed only very low signs of immune mediated destruction on day 30 posttranspant. Indeed, the arterial isografts treated by this protocol showed only very small inflammatory reaction in their wall as well. Contrary, aortal allografts

**Table 4. Percentage of fluorescence-labelled MHC class I and MHC class II antibody binding to Brown-Norway splenocyte in the presence of sera of Lewis recipients of aortal iso- or allografts.**

| | MHC class I | | MHC class II | |
|---|---|---|---|---|
| | Day 0 | Day 30 | Day 0 | Day 30 |
| ISOGRATS (CRYO-ISO, N = 6) | 111%±7% | 97%±5% | 90%±20% | 98%±7% |
| ALLOGRAFTS (CRYO-ALLO, N = 6) | 111%±22% | 47±19%* | 101±42% | 66±12% |

* P>0.05

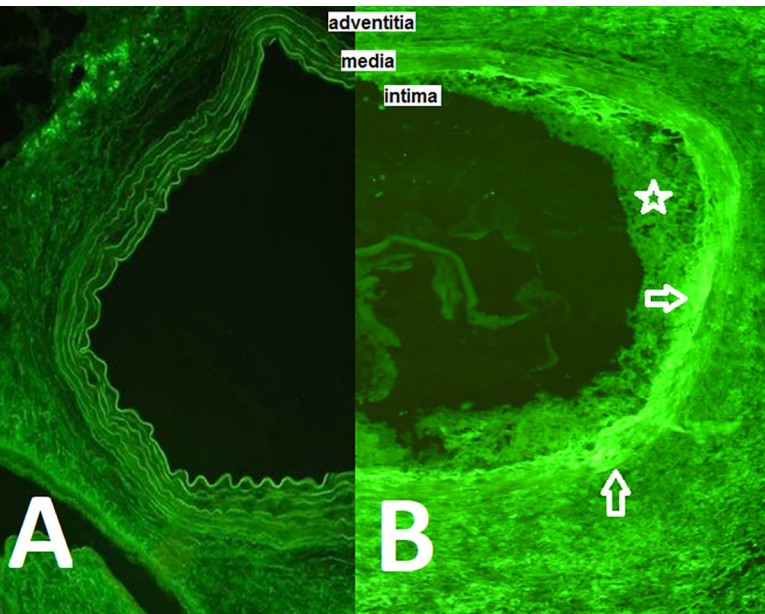

**Fig 2. Representative histological features of cryopreserved (A) and cold-stored aortal allografts (B) on day 30 posttransplant. Immunofluorescent staining for immunoglobulins G with a fluorescein isothiocyanate-conjugated antibody.** The rats aortal allografts processed in accordance with cryoconservation/slow thawing protocol used in the Czech national "Cryopreserved vascular grafts program" (A) showed clear differentiation of all three basic anatomical layers with no intimal hyperplasia and no IgG deposition in the medial layer. The rats aortal allografts processed in accordance with clinical cold-storage protocol (B) showed significant signs of rejection in all three wall layer represented by intimal hyperplasia (white star), destruction of medial layer with massive deposition of IgG (white arrows) and massive adventitial infiltration with immunocompetent cells of host origin. Original magnification x 100. adventitia–Tunica adventitia media–Tunica media intima–Tunica media.

treated by cold storage protocol showed statistically higher activation of recipient´s immune system with significant rejection changes in all three basic wall layers.

The most important issues of cryopreservation/thawing procedure during the pre-freezing period are both properties of conservation solution and duration of cold ischemia, respectively [10]. It was reported that prolonged cold-ischemia induced substantial damage to the arterial wall and that the endothelial cells are most susceptible to cold ischemia [16]. Indeed, Knight et al. have shown in rats, that 24 hours of cold ischemia induced vascular disease in both iso- and allografts at 4 weeks after transplantation [17].

In our experiment, pre-freezing ischemic times of aortal allografts were about 3–4 hours. Indeed, the cryopreserved grafts were stored immediately after recovery in pre-cooled organ preservation solution. It is very likely that this very short cold-ischemic time resulted both in significantly lower endothelial damage of cryopreserved grafts prior the transplantation and in minimal intimal reaction observed on day 30 posttransplant, respectively.

The fast warming at 37˚C in a water bath is mentioned in the most of recent published clinical works as the standard thawing method of cryopreserved arterial allografts [2]. However, recent experimental studies confirmed high damage level of rapidly thawed vascular tissues [18,19]. Contrary to this the slow thawing protocol diminished the immune response induced by cold-stored rat arterial allografts and improved their behavior after implantation [10]. However, this protocol is not suitable for clinical condition in operating room during the preparation of cryopreserved aortal grafts for implantation. First, the cryopreserved arterial grafts were transferred from storage tank to the freezer where they were thawed according to a slow thawing program to room temperature at a warming rate of 1˚C/minute. Second, the

cryoprotectant was removed gradually in tapered dilutions. According to our clinical protocol, the cassetes with cryopreserved grafts are thawed in a normal refrigerator within 2 hours and they are stored thereafter in the pre-cooled conservation solution until implantation [8].

Upregulation of major histocompatibility antigens expression by both ischemic and cryo-preservation/fast thawing injury with appearance of microfractures may increase the immuno-genicity of arterial allografts [18]. These antigens trigger strong donor specific anti-MHC Class I and anti-MHC Class II alloantibody production which drives the apoptosis of SMCs with medial layer destruction [20]. The medial layer of aortal allografts treated by our new cryocon-servation/slow thawing protocol showed no signs of destruction detected by light microscopy, no IgG deposition and no shrinkage compared to isografts 30 days posttransplant. However, in this work we didn´t use transmission electron microscopy to detect microfractures in aortal wall.

The statistically higher concentration of donor specific antibodies compared to preoperative values in recipients of cryopreserved allografts we observed only by anti MHC class I but not by anti MHC class II antibodies. MHC class II antigens are expressed on immunologically activated endothelial and smooth muscle cells [21]. So it is possible that our cryopreservation/ slow thawing protocol inhibited upregulation of major histocompatibility antigens expression in medial smooth muscle cells during the 30 day follow-up period. This hypothesis can be supported with no appearance of IgG clusters in medial layer of cryopreserved allografts on day 30 posttransplant as well. However, the effect of these serum anti MHC class I antibodies on immunological rejection of aortal allografts during the longer follow-up period is not clear.

The most significant limitation of this study is the comparison of current cryopreserved allografts data with historical cold-stored allografts data only. Recipients of cryografts were substantially smaller than the recipient of cold-stored allografts due to different ages of experimetal animals used in the two studies. However, the differences in the impact on aortal allograft rejection between both preservation protocols have been found to be much more pronounced than we expected. Therefore, further experiments with direct comparison of the two clinically used protocols are needed to confirm our observations.

## Conclusion

In conclusion, the present study considered that histological features of rat aortal allografts treated by our new cryopreservation/slow thawing clinical protocol did not show significant differences compared to isograft 30 day following transplantation. In addition, significantly lower immunogenicity of cryopreserved allografts compared to historical data of cold-stored allografts treated by clinical protocol was observed. However, other experiments are needed to confirm this effect. In the end, their positive results can influence the indication criteria and immunosuppressive therapy in the Czech national "Cryopreserved vascular grafts program" in the future.

## Supporting information

**S1 Table. The individual data per aortal graft of all measured time-periods of the cryocon-servation/slow thawing procedure.**
(XLSX)

## Author Contributions

**Conceptualization:** Rudolf Spunda, Pavel Mericka, Katrin Splith, Moritz Schmelzle, Felix Krenzien, Jaroslav Lindner, Miroslav Spacek, Ivan Matia.

**Data curation:** Jan Hruby, Rudolf Spunda, Pavel Mericka, Mikulas Mlcek, Katrin Splith, Moritz Schmelzle, Felix Krenzien, Miroslav Spacek, Ivan Matia.

**Formal analysis:** Jan Hruby, Ondrej Pecha, Moritz Schmelzle, Felix Krenzien, Miroslav Spacek, Ivan Matia.

**Funding acquisition:** Moritz Schmelzle, Jaroslav Lindner, Miroslav Spacek.

**Investigation:** Jan Hruby, Rudolf Spunda, Pavel Mericka, Felix Krenzien, Miroslav Spacek, Ivan Matia.

**Methodology:** Jan Hruby, Pavel Mericka, Mikulas Mlcek, Katrin Splith, Ivan Matia.

**Project administration:** Miroslav Spacek, Ivan Matia.

**Resources:** Mikulas Mlcek, Miroslav Spacek.

**Software:** Ondrej Pecha.

**Supervision:** Jaroslav Lindner, Ivan Matia.

**Validation:** Pavel Mericka, Ondrej Pecha, Ivan Matia.

**Writing – original draft:** Jan Hruby, Ivan Matia.

**Writing – review & editing:** Miroslav Spacek, Ivan Matia.

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
