## [Decision Letter · Decision Letter 0]

14 Jan 2020

PONE-D-19-35361

Influence of the New Standardized Clinical Cryopreservation/Slow Thawing Protocol on Immunogenicity of Arterial Allografts in Rats

PLOS ONE

Dear PD.Dr. Matia,

Thank you for submitting your manuscript to PLOS ONE. After careful consideration, we feel that it has merit but does not fully meet PLOS ONE’s publication criteria as it currently stands. Therefore, we invite you to submit a revised version of the manuscript that addresses the points raised during the review process.

The manuscript is potentially interesting, provided the authors are willing to improve it according to reviewers' suggestions.

We would appreciate receiving your revised manuscript by Feb 28 2020 11:59PM. To enhance the reproducibility of your results, we recommend that if applicable you deposit your laboratory protocols in protocols.io, where a protocol can be assigned its own identifier (DOI) such that it can be cited independently in the future. For instructions see: http://journals.plos.org/plosone/s/submission-guidelines#loc-laboratory-protocols

We look forward to receiving your revised manuscript.

Kind regards,

Prof. Raffaele Serra, M.D., Ph.D

Academic Editor

PLOS ONE

Journal Requirements:

2. To comply with PLOS ONE submission requirements, in your Methods section, please provide additional information regarding the experiments involving animals and ensure you have included details on (1) methods of sacrifice, (2) methods of anesthesia and/or analgesia, and (3) efforts to alleviate suffering.

Additional Editor Comments (if provided):

The manuscript is potentially interesting for the journal. Please see reviewers' suggestions and amend your manuscript accordingly.

Reviewers' comments:

Reviewer's Responses to Questions

**Comments to the Author**

1. Is the manuscript technically sound, and do the data support the conclusions?

Reviewer #1: Partly

2. Has the statistical analysis been performed appropriately and rigorously? 

Reviewer #1: Yes

3. Have the authors made all data underlying the findings in their manuscript fully available?

Reviewer #1: No

4. Is the manuscript presented in an intelligible fashion and written in standard English?

Reviewer #1: Yes

5. Review Comments to the Author

Reviewer #1: The overall goal of this study aims to determine if a new cryopreservation and slow thawing protocol for aortal grafts elicits an acute immune response and to compare this response to the current standard "cold-storage" protocol. Given the urgent clinical need for additional allografts for patients on the waiting list, the rationale for this study is excellent and provides valuable data toward the development of a protocol to be tested in humans. The results suggest that their new protocol for storing allografts prior to use reduces immunogenicity which could lead to improved functional results.

However, the comparison of the CRYO groups to the COLD groups needs some additional description and comparison to ensure that their conclusion is accurate. First, the authors state in the introduction that protocols used in different center vary considerably and can influence the end results of the allografts. In this regard, Table 1 aims to compare the key steps in the protocols between the CRYO and COLD protocols. This table needs to have more details included or the details need to be added to the body of the manuscript. First, the units of time are not included (minutes, hours, etc.) other than the time recorded in days. Second, since there are only a few rats in each group, the individual data per rat should be included in the manuscript, either as a separate table or in supplemental material. This could be especially important for the time periods that have a wide range of results such as the pre-freezing CIT. The range for CRYO-ALLO is between 1:23 and 4:45 which could influence the final quality and would need to be further refined in developing a protocol to translate to humans. Similarly, post-thawing also has a similarly wide range of results. Were the animals with the shortest or longest times the same in both the pre-freezing and post-thawing? Is there any correlation to the histological and immunohistological results presented in Table 3 with the total time of the procedure presented in Table 1? Was there an expected time for each time-period of the cryopreservation and thawing process and any plan for inclusion/exclusion of results if the expected time was exceeded?

It appears that the results for the COLD protocol were not collected as part of this study but results from this study compared to results previously published in 2007. This comment is based on the the sentence that begins on page 7, line 134. What controls were used to ensure that the surgical procedures used to collect the tissues and graft the procedures were comparable between studies performed many years apart? In addition, it appears that the CRYO rats were substantially smaller than the COLD rats, presumably due to different ages being used in the two studies. Age should be included in the results and any implications of age related responses should be discussed in the manuscript.

It needs to be made very clear if the COLD results being used to compare the CRYO results are previously published or replicated in the current study. The results section indicates that 24 aortal grafts were transplanted which suggests that all were done as part of the current study. However, there were two COLD animals that died, similar to the results in the 2007 published paper. The PLOS ONE instructions clearly states that results that have been previously published, in whole or in part, are not accepted, so it is critical to be clear if the COLD results are new or previously published.

There are two sections in the results labelled "Histology and immunohistology of cryopreserved and cold-store aortal allografts on day 30 posttransplant" so some reorganization is needed for clarity. The histological results in Figure 1 need to have a figure legend included to orient the reader to the images and interpretation of the results. In addition, a comparison between CRYO-ALLO and CRYO-ISO would complement these results well. The methods indicate that both intimal and medial thickness were measured (page 13, line 251-252) but only medial thickness was reported in table 4. Intimal thickness should be presented.

Finally, there are some typos that should be corrected (e.g. immunoglobulines, nitrogene). A proof read for standard English is also suggested.

6. PLOS authors have the option to publish the peer review history of their article (what does this mean?). If published, this will include your full peer review and any attached files.

Reviewer #1: No

---

## [Author Response · Author response to Decision Letter 0]

30 Jan 2020

Dear Reviewer of PLOSone,

I would like to send our responses to your review of our Article: Influence of the New Standardized Clinical Cryopreservation/Slow Thawing Protocol on Immunogenicity of Arterial Allografts in Rats

I would like to thank you for your review comments that helped us to improve our publication.

Yours faithfully

Ivan Matia

5. Review Comments to the Author

Reviewer #1: The overall goal of this study aims to determine if a new cryopreservation and slow thawing protocol for aortal grafts elicits an acute immune response and to compare this response to the current standard "cold-storage" protocol. Given the urgent clinical need for additional allografts for patients on the waiting list, the rationale for this study is excellent and provides valuable data toward the development of a protocol to be tested in humans. The results suggest that their new protocol for storing allografts prior to use reduces immunogenicity which could lead to improved functional results.

However, the comparison of the CRYO groups to the COLD groups needs some additional description and comparison to ensure that their conclusion is accurate. First, the authors state in the introduction that protocols used in different center vary considerably and can influence the end results of the allografts. In this regard, Table 1 aims to compare the key steps in the protocols between the CRYO and COLD protocols. This table needs to have more details included or the details need to be added to the body of the manuscript. First, the units of time are not included (minutes, hours, etc.) other than the time recorded in days. 

Units were added into the Table 1

Table 1. Time-periods of the cryoconservation/slow thawing procedure.

Second, since there are only a few rats in each group, the individual data per rat should be included in the manuscript, either as a separate table or in supplemental material. 

The individual data per rat were added in suplemental material.

This could be especially important for the time periods that have a wide range of results such as the pre-freezing CIT. The range for CRYO-ALLO is between 1:23 and 4:45 which could influence the final quality and would need to be further refined in developing a protocol to translate to humans. Similarly, post-thawing also has a similarly wide range of results. Were the animals with the shortest or longest times the same in both the pre-freezing and post-thawing?

The total cold ischemic time of cryopreserved aortal grafts (pre-freezing + post-thawing cold ischemic time) was 313±62 minutes for isografts and 504±198 minutes for allografts, respectively. No statistical difference was observed between both groups.

This information was added to the manuscript.

Is there any correlation to the histological and immunohistological results presented in Table 3 with the total time of the procedure presented in Table 1? 

No, we have seen no correlation to the total time oft he procedure in Table 1. 

Was there an expected time for each time-period of the cryopreservation and thawing process and any plan for inclusion/exclusion of results if the expected time was exceeded?

No there was no expected time for any time-period. 

All steps and time intervals of presented experimental protocol are in accordance with our clinical protocol. However, times of pre-freezing and post-thawing cold ischemia vary widely in clinical conditions. In this experiment we achieved shortening of both cold ischemic times.

Špaček M, Měřička P, Janoušek L, Štádler P, Adamec M, Vlachovský R, et al. Current vascular allograft procurement, cryopreservation and transplantation techniques in the Czech Republic. Adv Clin Exp Med. 2019;28: 529–534. 

Moreover the information about total ischemic time in both groups was added as follows:

The total cold ischemic time of cryopreserved aortal grafts (pre-freezing + post-thawing cold ischemic time) was 313±62 minutes for isografts and 504±198 minutes for allografts, respectively. No statistical difference was observed between both groups.

It appears that the results for the COLD protocol were not collected as part of this study but results from this study compared to results previously published in 2007. This comment is based on the the sentence that begins on page 7, line 134. What controls were used to ensure that the surgical procedures used to collect the tissues and graft the procedures were comparable between studies performed many years apart?

Yes the results for the COLD protocol were collected as part of the previously published study. This information is newly mentioned in the text as follows:

The original core data of recent experiment were compared to historical data of our cold-stored allografts experiment.[13] 

The cryopreserved aortal grafts in recent experiment were transplanted by the same anesthesiological and surgical techniques used previously in our cold-stored aortal grafts experiment. Both experiments were supervised by the same principal investigator (I.M.). 

Moreover, the third aim of our study was changed as follows:

3. to compare the data of current cryopreservation/slow thawing experiment to data obtained previously by our cold-storage rat experiment to study the differences in immunogenicity of both cryopreserved and cold-stored aortal allografts, respectively 

In the new version of our manuscript, there are no numerical data published previously. 

In addition, it appears that the CRYO rats were substantially smaller than the COLD rats, presumably due to different ages being used in the two studies. Age should be included in the results and any implications of age related responses should be discussed in the manuscript.

The limitation section was added to he manuscript as follows:

The most significant limitation of this study is the comparison of current cryopreserved allografts data with historical cold-stored allografts data only. Recipients of cryografts were substantially smaller than the recipient of cold-stored allografts due to different ages being used in our two studies. However, the differences in the impact on aortal allograft rejection between both preservation protocols have been found to be much more pronounced than we expected. Therefore, further experiments with direct comparison of the two clinically used protocols are needed to confirm our observations.

It needs to be made very clear if the COLD results being used to compare the CRYO results are previously published or replicated in the current study. The results section indicates that 24 aortal grafts were transplanted which suggests that all were done as part of the current study. However, there were two COLD animals that died, similar to the results in the 2007 published paper. The PLOS ONE instructions clearly states that results that have been previously published, in whole or in part, are not accepted, so it is critical to be clear if the COLD results are new or previously published.

The information on the COLD results has been extended and explaned in the text as mentioned above. 

There are two sections in the results labelled "Histology and immunohistology of cryopreserved and cold-store aortal allografts on day 30 posttransplant" so some reorganization is needed for clarity. 

The histology section was reorganized as follows:

Histology and immunohistology of cryopreserved isografts and allografts on day 30 posttransplant

The histological results in Figure 1 need to have a figure legend included to orient the reader to the images and interpretation of the results.

The new Figure 1 composed of six histological images was added:

Representative light microscopic histological features of both isografts (a,b,c) and allografts (d,e,f) treated by our new cryoconservation/slow thawing protocol and obtained on day 30 following transplantation.

The previous Figure 1 is now listed as Figure 2.The captions were added to the image.

In addition, a comparison between CRYO-ALLO and CRYO-ISO would complement these results well. The methods indicate that both intimal and medial thickness were measured (page 13, line 251-252) but only medial thickness was reported in table 4. Intimal thickness should be presented.

Table 3 compares now the results between CRYO-ALLO and CRYO-ISO group 

Table 3. Histological and immunohistological parameters of cryopreserved aortal grafts under study on day 30 posttransplant.

Finally, there are some typos that should be corrected (e.g. immunoglobulines, nitrogene). A proof read for standard English is also suggested.

Corrected.

---

## [Decision Letter · Decision Letter 1]

26 Feb 2020

Influence of the New Standardized Clinical Cryopreservation/Slow Thawing Protocol on Immunogenicity of Arterial Allografts in Rats

PONE-D-19-35361R1

Dear Dr. Matia,

We are pleased to inform you that your manuscript has been judged scientifically suitable for publication and will be formally accepted for publication once it complies with all outstanding technical requirements.

With kind regards,

Prof. Raffaele Serra, M.D., Ph.D

Academic Editor

PLOS ONE

Additional Editor Comments (optional):

amended manuscript is acceptable

Reviewers' comments:

Reviewer's Responses to Questions

**Comments to the Author**

1. If the authors have adequately addressed your comments raised in a previous round of review and you feel that this manuscript is now acceptable for publication, you may indicate that here to bypass the “Comments to the Author” section, enter your conflict of interest statement in the “Confidential to Editor” section, and submit your "Accept" recommendation.

Reviewer #1: All comments have been addressed

2. Is the manuscript technically sound, and do the data support the conclusions?

Reviewer #1: Yes

3. Has the statistical analysis been performed appropriately and rigorously? 

Reviewer #1: Yes

4. Have the authors made all data underlying the findings in their manuscript fully available?

Reviewer #1: Yes

5. Is the manuscript presented in an intelligible fashion and written in standard English?

Reviewer #1: Yes

6. Review Comments to the Author

Reviewer #1: Most of the previous comments have been adequately addressed.

However, there is still a minor concern related to the time of each step and the potential impact on the overall results. The authors claim that there are no statistical differences but the number of samples might limit the finding of any statistical differences. The pre-freezing time, the post-thawing time, and the total time are all greater in the ALLO group compared to the ISO group. Are the post-thawing times statistically different between groups? It would be ideal to confirm that all aspects of the cryo protocol have been compared appropriately.

7. PLOS authors have the option to publish the peer review history of their article (what does this mean?). If published, this will include your full peer review and any attached files.

Reviewer #1: No

---

## [Editor Report · Acceptance letter]

2 Mar 2020

PONE-D-19-35361R1 

Influence of the New Standardized Clinical Cryopreservation/Slow Thawing Protocol on Immunogenicity of Arterial Allografts in Rats 

Dear Dr. Matia:

I am pleased to inform you that your manuscript has been deemed suitable for publication in PLOS ONE. Congratulations! Your manuscript is now with our production department. 

With kind regards,

on behalf of

Prof. Raffaele Serra 

Academic Editor

PLOS ONE